# Influence of Aging Time on Vertical Static Stiffness of Air Spring

Zhaijun Lu [1,2,3], Penghao Si [1,2,3], Hao Xiao [1,2,3] and Jiefu Liu [1,2,3,*]

1    School of Traffic & Transportation Engineering, Central South University, Changsha 410017, China; qlzjzd@csu.edu.cn (Z.L.); sipenghao2022@163.com (P.S.); xiaohao03172022@163.com (H.X.)
2    Key Laboratory of Traffic Safety on Track, Ministry of Education, Changsha 410075, China
3    National & Local Joint Engineering Research Center of Safety Technology for Rail Vehicle, Changsha 410075, China
*    Correspondence: liujiefu@csu.edu.cn

**Abstract:** To study the aging mechanism of air springs, the effect of aging time on the vertical static stiffness of an air spring was systematically analyzed by means of an accelerated aging test and finite element simulation. Accelerated aging tests were carried out on the entire air spring, rubber material, and cord material, and the vertical static stiffness and elastic moduli of the rubber and cord materials of the entire air spring were obtained with aging time. The finite element simulation model of the air spring was established. Based on the experimental data, the influences of the elastic moduli of the rubber and cord materials, aged for different times, and the cord angle on the vertical static stiffness of an air spring were simulated and analyzed, and the law of the influence of aging on the vertical static stiffness characteristics of air springs was revealed.

**Keywords:** air spring; finite element; aging; stiffness characteristics; Abaqus

## 1. Introduction

Air springs are one of the most important components in the secondary suspension system of rail vehicles, and their performance directly affects the stability and safety of those vehicles. Exploring the application of a new prediction theory and method in predicting the performance decline of the train air spring can provide a theoretical basis and reference for detecting the performance of an air spring, so that it can be repaired or replaced. Research on, and applications of, air springs started earlier outside China [1,2]. The rubber spring was simplified and equivalent to parallel between the linear spring and linear viscoelastic damping, and the mechanical characteristics of the rubber spring were studied using a rubber spring experimental platform [3,4]. The air spring system has been studied for equivalent mechanical modeling, forming three air spring equivalent mechanical models: the Nishimura, VAMPIRE, and Berg models [5,6]. Considering the thermodynamic properties of the air spring pneumatic system, the effective friction of the connecting line, and the viscoelastic damping of the rubber airbag, the dynamic model of the air spring system was established. It can accurately predict the amplitude and frequency correlation characteristics and dynamic characteristics of the air spring [7].

According to the traditional mechanical air spring model, the static and dynamic mathematical models of the air spring were established [8]. The nonlinear dynamic stiffness and static stiffness models of the cystic air spring were developed based on the thermal dynamics and by using an improved analytical calculation method. The finite element analysis of air springs is mainly based on Abaqus, MSC, Marc, ADAMS, and other software [9–12]. Wang et al. established the air spring model through the Abaqus analysis software. They used the fluid analysis software STAR-CCM to simulate the internal gas, exchanging data between the two through SIMULIA to calculate the dynamic stiffness of the air spring [13]. Yu studied the relevant mechanical properties of air springs under low temperature, improving the air spring performance at low temperatures [14]. The service

life of the air spring was investigated by an airbag rubber aging test, and the physical performance of the rubber was studied [15]. In order to accurately obtain the mechanical characteristics of air springs, it is necessary to select the finite element analysis software that can effectively deal with all kinds of nonlinear problems and various element types to simulate air springs. The Abaqus finite element simulation analysis software can simulate nonlinear problems well [16–20].

Past research on the mechanical characteristics of air springs has considered the influence of aging less. In fact, some scholars have studied the effect of aging time on concrete [21]. However, because of the constant exposure of air springs to the atmosphere during service, the effect of sunlight on airbags, acid–alkali liquid corrosion, and other natural aging factors, the parameters of the air spring airbag cord and rubber change, affecting the mechanical characteristics of the air spring, and the comfort and safety of vehicle operation. In this study, the vertical stiffness of the air spring and the elastic moduli at different angles were investigated. A theoretical basis for air spring structure design and air spring maintenance in practical applications is provided.

## 2. Air Spring Device

As shown in Figure 1, the air spring is composed of a buckle, an upper cover plate, an air bag, a wear plate, a support plate, and an auxiliary spring. Among them, the air spring rubber is made of material with good air tightness. The cord layers in the middle of the inner and outer rubber layers are arranged in a cross.

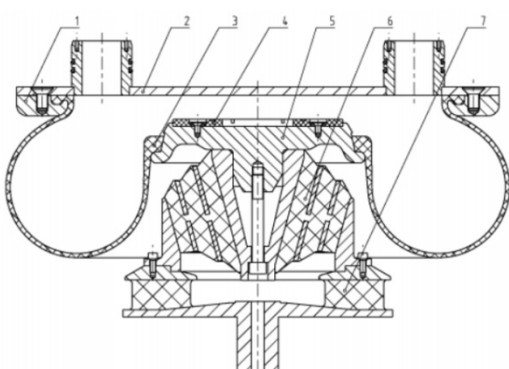

1-buckle  2-upper cover  3-airbag  4-wear plate  5-support plate  6，7-auxiliary spring

**Figure 1.** Air spring components.

The auxiliary spring used in this type of air spring is composed of a conical rubber stack and a layered spring. The auxiliary spring and the air spring capsule are connected in series with each other, which can better improve the dynamic performance of the air spring. The auxiliary spring plays the role of emergency vibration damping and load bearing in the event of an accidental failure of the air spring, such as deflation, which improves the safety performance of the air spring.

## 3. Experimental Study

### 3.1. Aging Test Method

To study the vertical static stiffness change of the air spring under aging, a hot-air accelerated aging test was used to simulate the natural environmental aging of the air spring. First, the overall trend of vertical static stiffness increasing with aging time was studied using an air spring component aging test. Then, the influence that the rubber material and curtain material had on the air spring static characteristics was studied by a material accelerated aging test. The specific test scheme is as follows.

An air spring was installed in an aging box, and the temperature was adjusted to 100 °C for 1, 2, 4, 8, and 12 days. After reaching the set aging time, the air spring was removed and set for 16 h at room temperature before the vertical static stiffness test was

conducted. According to the relevant test standards of TB/T28412010 railway vehicle air springs, the stiffness of the air spring was tested using the air spring stiffness tester.

The air spring rubber and cord materials were laid out and cut into "dumbbell" shapes, test equipment, and test pieces, as shown in Figure 2. The aging box was heated to 100 °C and insulated. The rubber test pieces were placed in the aging box and removed when the insulation time reached 1, 2, 4, 8, and 12 days. Three tests were prepared for each aging condition. After the set aging time was reached, each specimen was removed, the Young's modulus was cooled at room temperature for 12–24 h, and tensile tests were performed on the cord and rubber samples, according to GB/T 528-2009.

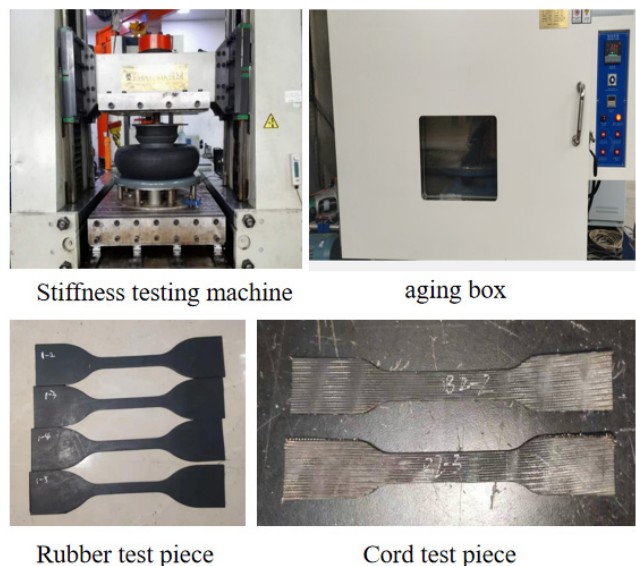

**Figure 2.** Test equipment and test pieces.

### 3.2. Aging Test Results and Analysis

The curve of the air spring with aging time is shown in Figure 3. During the initial inflation, the internal pressure of the air spring fluctuated, and the measured internal pressure was 0.612 MPa. As the figure shows, the vertical static stiffness of the air spring became larger with the increase in aging time. The vertical static stiffness of the air spring was 251.5 N/mm before aging, and it was 277.4 N/mm after aging—an increase of 10.3%.

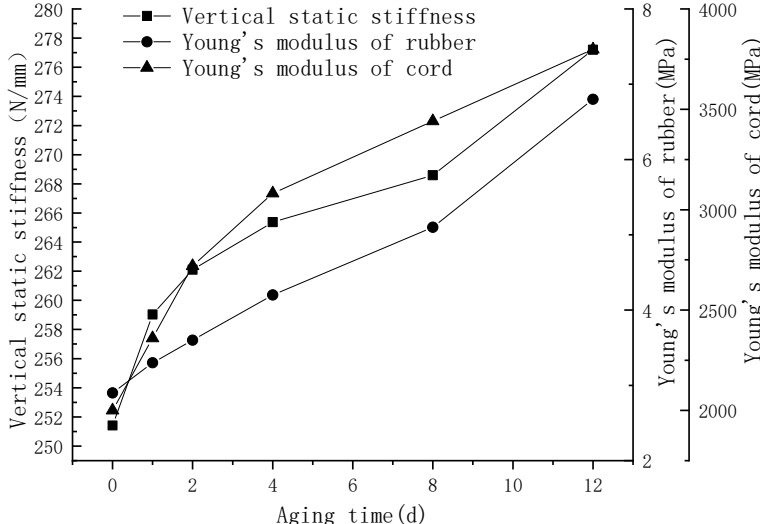

**Figure 3.** Aging test results.

The curves of the Young's moduli of the rubber material and curtain material are shown in Figure 2. The tensile test of the rubber materials and cord materials shall be carried out according to the standard GB/T 528-2009. In the actual application process, the airbag rubber was constrained by the curtain cord, and its strain was less than 10% [22]. It can be regarded as line elastic, so the Young's moduli of the rubber and curtain line were obtained by fitting the stress–strain data in the range of 1–10% strain. The figure shows that the initial Young's modulus of the rubber was 2.9 MPa, and it increased with the increase in thermal oxygen aging time. It increased by 131.1% after 12 days of aging, which indicates that the rubber hardened under such aging conditions. This is because the rubber material produces thermal aging and crosslink reactions of molecular chain degradation; thus, as the molecular chain degradation dominates, the rubber softens, eventually becoming soft and sticky. If the crosslink reaction dominates, the rubber eventually becomes brittle and hard. The test results reveal that the rubber material mainly underwent a crosslinking reaction. As shown in Figure 2, the initial Young's modulus of the cord was 2000 MPa. It became larger with aging time and was 89.9% after 12 days. This was because of the thermoplasticity of the curtain line, which changes in crystallinity in high-temperature aging [23], causing the Young's modulus to increase.

### 4. Numerical Simulation

The finite element model of the air spring was established using Abaqus software. Since the structure of the air spring and the load it bears are symmetric, only 1/2 of the air spring model was built for the simulation, to simplify the calculation. The rubber airbag thickness of the air spring is 6 mm and can be regarded as a thin wall structure, simplified to a shell structure, and discrete using a four-node shell unit S4R; a rebar unit can be used to simulate the curtain of the air spring. The upper and lower cover plates are made of metal materials. They are set as analytical rigid bodies, and the attributes of the mesh are given as r3d3 and r3d4. The conical emergency spring is mainly composed of a layer spring and conical spring, through step coordination and screw material. The emergency spring is regarded as a whole part during modeling. When the modeling is complete, the metal and rubber parts are divided into areas, and different material attributes are given to each segmentation area. The grid division is shown in Figure 4. The cord ply of the air spring is located in the middle layer of the airbag, and is wrapped by two layers of rubber material inside and outside. The main function of the cord layer is to restrain the deformation of the air spring rubber airbag, and the paving form is cross arranged. In the Abaqus finite element simulation software, the rebar element model can be used to simulate this structure. The parameters that the rebar model can define include the Young's modulus of rubber, area of the cord, cord spacing, cord layout angle, etc.

The symmetry constraints are set at the symmetric surface of the model. The reference point at the center of the upper cover surface is defined to establish coupling constraints with the upper cover surface. The auxiliary spring base is completely secured. Abaqus is used to simulate the static mechanical characteristics of the air spring. Abaqus/Standard is used for the calculation analysis, and the simulation process is divided into two analysis steps. The first is to simulate the air filling process of the capsule. First, fixed constraints are placed on the upper cover plate and the auxiliary spring reference point, and then gas is filled into the capsule to compare the simulation results with the test results, so that the maximum air pressure is 0.612 MPa in the actual test. The second analysis step simulates the process of loading of external force. The constraint of the auxiliary spring base is maintained, the vertical constraint of the upper cover plate is released, and the vertical displacement load is applied to the upper cover plate. Through two-step simulation, the reaction and displacement at the reference point can reveal the vertical stiffness of the air spring.

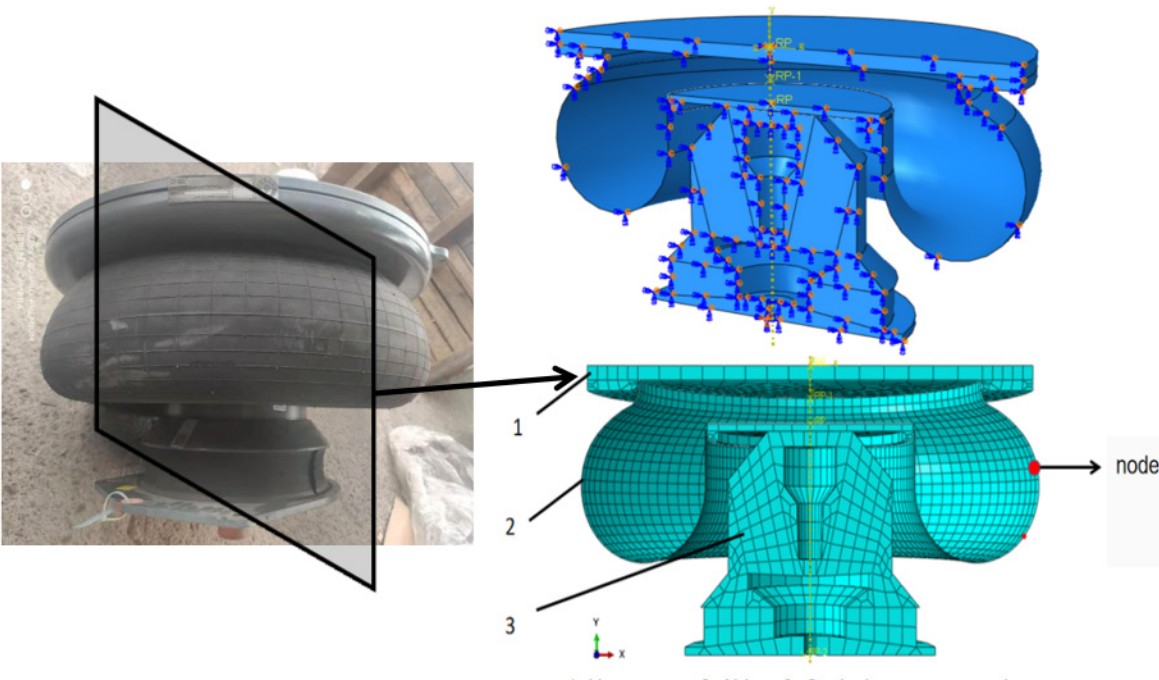

1. Upper cover 2. Airbag 3. Conical emergency spring

**Figure 4.** Finite element model of air spring.

The rubber and curtain parameters obtained with the different aging times were fed into the air spring finite element model, and the air spring vertical static stiffness was obtained, as shown in Figure 5. As the figure reveals, aging increases the air spring vertical stiffness. The 12-day increase was from 249.1 to 289.7 N/mm, and the relative error was between 251.5 N/mm and 277.4 N/mm. Therefore, the simulation analysis model and simulation results are reasonable and credible.

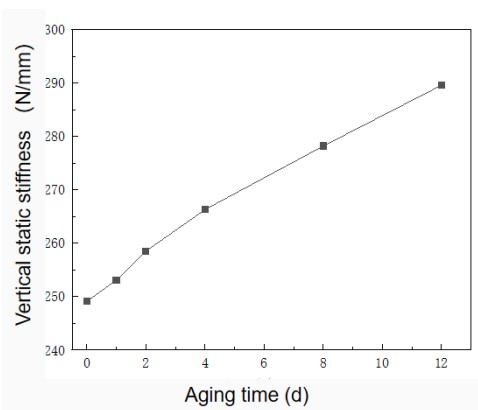

**Figure 5.** Variation curve of vertical static stiffness of air spring with aging time.

The vertical static stiffness obtained from the simulations and experiments increased by 16.3% and 10.3% before and after aging, respectively. The vertical static stiffness of the air spring increased faster than that obtained in the experiment. The main reason for this phenomenon is that, compared with the air spring in the middle of the rubber, the rubber and curtain materials are more fully exposed to the hot-air environment, and the rubber and cord material, under the same aging conditions, using the material parameters of the spring aging results, have a faster aging rate.

## 5. Simulation on Vertical Static Stiffness of Air Springs with Aging Time

### 5.1. Effects of Cord Aging

The simulation analysis results of the air spring vertical static stiffness, only considering rubber aging, are shown in Figure 6, and the static stiffness after 12 days of aging increased by 11.9% compared with that before aging. As the figure shows, as the rubber ages, the vertical static stiffness of the air spring decreases after one day of aging, and it increases between 1 and 12 days.

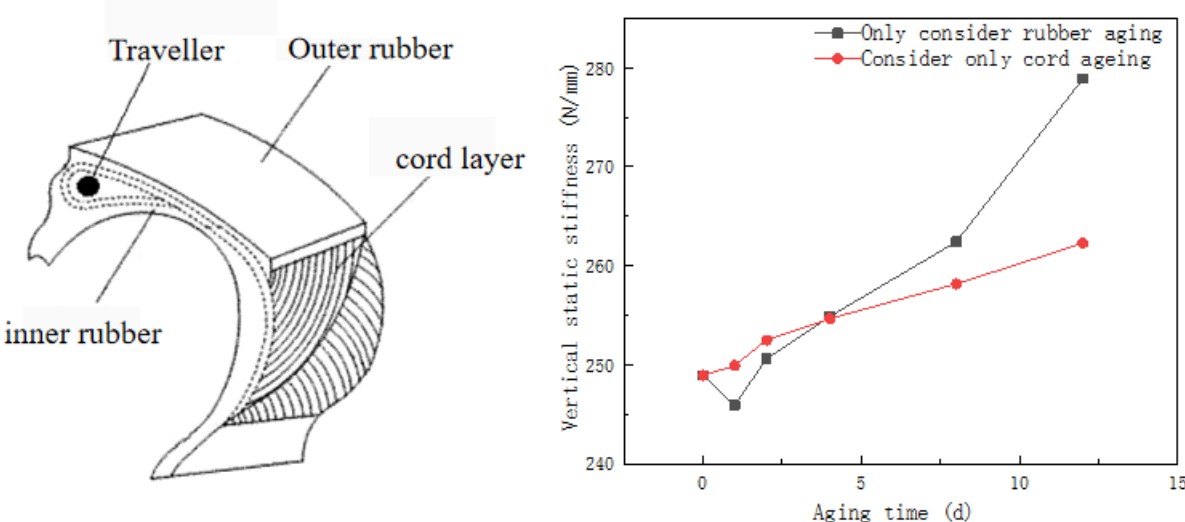

**Figure 6.** Consider the vertical static stiffness of an air spring affected by rubber aging and cord aging only.

The effective area of the air spring is the theoretical equivalent area [24]. The effective bearing area of the air spring is the effective area, which refers to the bearing area that can just offset the current vertical load. It should be emphasized that the effective area of the air spring is not the maximum cross-sectional area of the airbag or the contact area with the cover plate, because there is always uniformly distributed tension in the longitudinal section of the rubber airbag, and some pressure bearing areas should be used to offset the tension of the airbag itself. The greater the effective area, the greater the stiffness [25]. The vertical static stiffness of the air spring is mainly proportional to the air pressure and effective area. As the aging time increases, the rubber becomes hard and brittle. When subjected to a vertical displacement load of the same size, the deformation of the air spring airbag is smaller, making the effective area of the air spring decrease and the air pressure in the airbag increase. The effective sectional area reduces the stiffness of the air spring, and the air pressure increases the stiffness. When the effect of the effective area is greater than the effect of increasing pressure, the vertical static stiffness of the air spring decreases; when the effective area is less than the increasing pressure, the vertical static stiffness of the air spring increases. According to the above analysis, the effective area plays the main role when aged up to one day; in 1–12 days, air pressure plays the main role.

### 5.2. Effects of Rubber Aging

The simulation analysis results of the air spring vertical static stiffness, only considering the curtain aging, are shown in Figure 6. The figure shows that the gradually aging curtain line gradually increases the air spring vertical static stiffness, which increased by 5.3% at 12 days compared with that before aging.

The vertical static stiffness of the air spring is proportional to its effective area, so one can compare the lateral displacement on the airbag node, the lateral displacement of the air spring, and the larger effective area. The curve of the lateral displacement of the node on the air spring airbag, shown in Figure 5, with the analysis step time is shown in Figure 7. As Figure 7a shows, when considering rubber aging only, the lateral displacement

of the node during the first load step inflation increases; then, the air pressure decreases because the node moves inward at this position, with the deformation of the airbag. With the increase in aging time, the lateral displacement of the node on the airbag decreases significantly—that is, the effective bearing area of the air spring decreases. As shown in Figure 7b, the lateral displacement of the node on the airbag increases slightly. In other words, the effective bearing area of the air spring becomes larger, but the change amount is less obvious than when only rubber aging is considered.

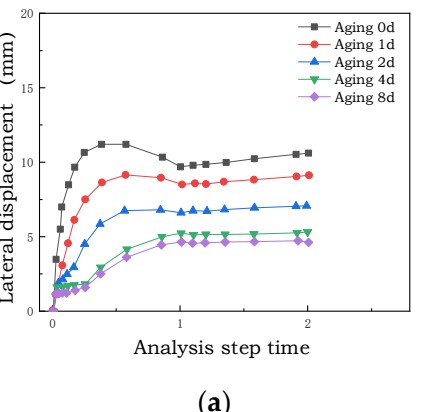
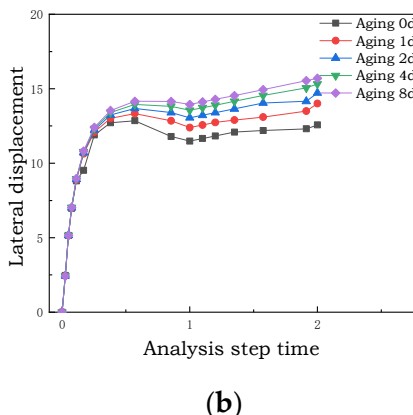

(**a**)　　　　　　　　　　　　　　　　　　　　　　(**b**)

**Figure 7.** The lateral displacement of the joint varies with the time of the analysis step: (**a**) considering the rubber aging, the lateral displacement of the node varies with the analysis step time; (**b**) considering the aging of the cord, the lateral displacement of the node varies with the analysis step time.

The vertical stiffness of the air spring and the internal pressure of the airbag are proportional, and the internal pressure of the airbag volume is inversely proportional to the change in the shape of the airbag, which directly reflects the change in the airbag volume, so the shapes of the airbag before and after aging can be compared. Figure 8 shows a diagram comparing the shapes of the air bag cross-section before and after aging. As Figure 8a shows, when considering the rubber aging only, the airbag volume after 12 days of aging is significantly smaller, causing a large internal pressure, and the impact of the large internal pressure is greater than the impact of a smaller effective area, so the empty spring vertical to the static stiffness increases. Compared with the shape change of the airbag before and after rubber aging, it can be observed from Figure 8b that the airbag section area change after curtain aging is relatively small, so the airbag volume change is also small; thus, the internal pressure change of the airbag is also smaller. Figure 8 shows the curve of the maximum air pressure during loading time under different conditions. As Figure 9 shows, when considering the curtain aging only, the internal pressure increase of the airbag is less than the air pressure increase of the rubber aging, so the impact of curtain aging on the air spring stiffness is less than that of rubber aging. In summary, it is considered that the effective area reduces the vertical static stiffness, the internal pressure of the airbag increases, and the internal pressure of the airbag plays a leading role. Therefore, the air spring gradually increases, but the increase is less than the value of rubber aging.

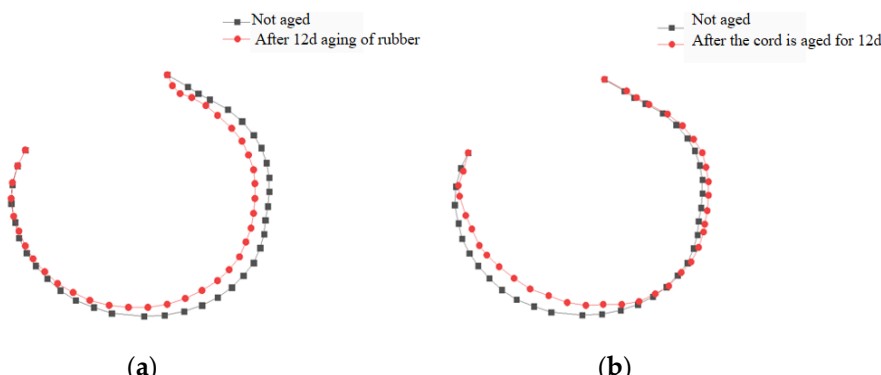

**Figure 8.** Schematic diagram of airbag section shape change: (**a**) considering the aging of rubber, the shape of the airbag section changes; (**b**) considering the aging of the cord, the airbag cross-sectional shape changes.

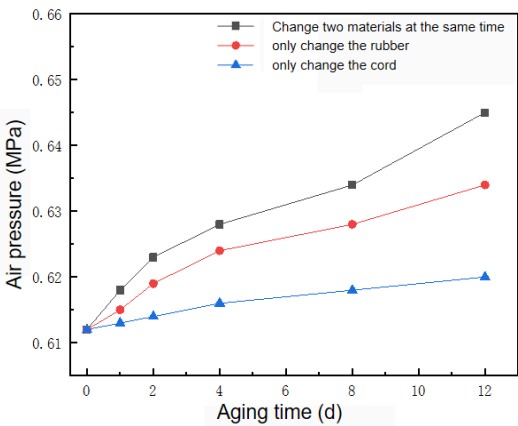

**Figure 9.** Comparison of air pressure with aging time.

### 5.3. Influence of Cord Angle on Vertical Static Stiffness of Air Spring during Aging

To examine the vertical stiffnesses of air springs with different curtain angles, affected by aging time, the vertical stiffnesses of air springs with 20°, 30°, 40°, and 50° angles were analyzed. The simulation analysis results are shown in Figure 10. The air spring stiffness decreases because of the curtain angle, and the trend is not affected by aging. When the curtain angle is 25° and the air spring vertical stiffness is minimal because of aging, the air spring at that angle is relatively stable.

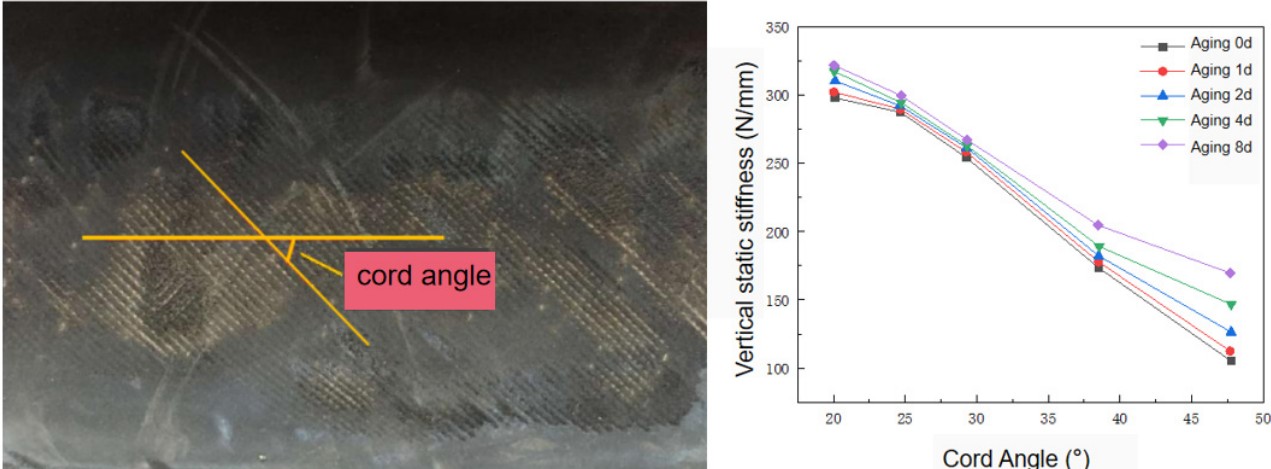

**Figure 10.** Vertical static stiffness of air spring with different cord angles during aging.

*5.4. Influence of Conical Emergency Spring on Vertical Static Stiffness of Air Spring during Aging*

An emergency rubber pile, also known as an "emergency spring," has, as its main materials, rubber and steel gaskets. The elastic modulus of steel does not change and is negligible, but the rubber material is greatly affected by aging; thus, it is necessary to study the vertical static stiffness of the air spring when there is no emergency spring. The conical emergency rubber stack portion in the finite element model was removed, and all six degrees of freedom of the lower cover reference point were constrained.

Figure 11 is a contrast diagram of the springs during aging. As shown in the figure, the vertical static stiffness of the conical emergency spring is less than that without an emergency spring, and the vertical static stiffness of the spring increases with the aging time. This indicates that the conical emergency spring helps attenuate the impact of aging on the vertical static stiffness of the air spring.

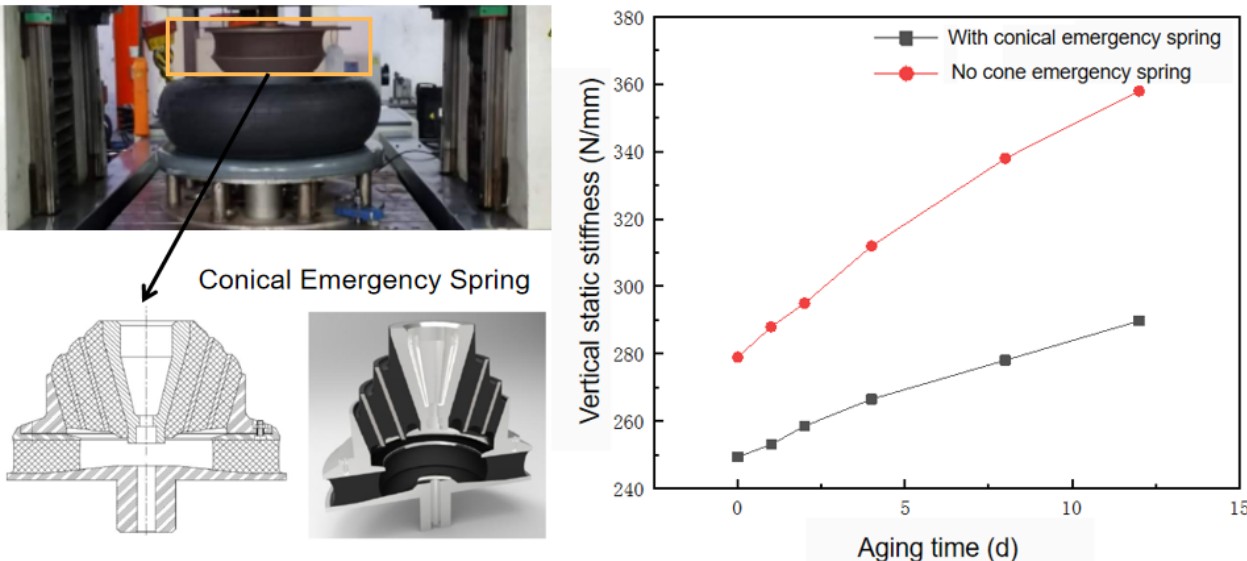

**Figure 11.** Vertical static stiffness of air spring with and without conical emergency spring during aging.

## 6. Conclusions

The vertical static stiffness of the air spring during aging was calculated by Abaqus finite element software. The effects of different parameters on the stiffness characteristics of the air spring during aging were compared, and the law of each parameter affecting the stiffness characteristics of the air spring during aging was analyzed. Based on the above analysis, the following conclusions are drawn.

(1) The vertical static stiffness of the air spring increases with the aging time. Both the aging of the air spring cord material and that of the rubber material lead to an increase in the vertical static stiffness, with the latter having a greater effect. Separately analyzing the effects of the rubber and cord materials on the vertical static stiffness of the air spring under aging reveals that, when only the rubber parameters are changed, the vertical static stiffness of the air spring first decreases and then increases with the increase in aging time. However, when only the parameters of the cord material are changed, the vertical static stiffness of the air spring increases with the aging time. The reasons for this phenomenon can be analyzed by studying the change in the effective area and the change in the internal pressure of the airbag.

(2) From a structural point of view, the vertical static stiffness of the air spring decreases with the increase in the cord angle, and this trend is not affected by aging. The stiffness characteristics are more stable. The vertical static stiffness of the air spring with an emergency spring increases at a smaller rate with aging time, indicating that the conical emergency spring helps to reduce the effect of aging on the vertical static stiffness of the air spring.

**Author Contributions:** Conceptualization, Z.L.; Data curation, H.X.; Writing—original draft, P.S.; Writing—review & editing, J.L. All authors have read and agreed to the published version of the manuscript.

**Funding:** This research received no external funding.

**Institutional Review Board Statement:** Not applicable.

**Informed Consent Statement:** Not applicable.

**Data Availability Statement:** Not applicable.

**Acknowledgments:** This work was supported by the National Key Research and Development Plan (2018YFB1201700). The authors would like to express their gratitude.

**Conflicts of Interest:** The authors declare no conflict of interest.

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
