# Peer review of "Influence of Aging Time on Vertical Static Stiffness of Air Spring"

_applsci, doi:10.3390/app12094219_

Round 1

Reviewer 1 Report

The manuscript submitted for publication in Applied Sciences is dedicated to the study of the influence of aging time on vertical stiffness of air spring. The authors developed  numerical simulations of air sprint stiffness calculation, and also performed experimental study. They used a commercial software, ABAQUS.

The topic is interesting and useful for industry, but need more updates as below.

1) So far as experimentation, I don't see any standards used for a couple of experiments. Please address test standards for your test. More details are necessary for other researchers

2) in Fig 3. Need labeling for the Young's modulus axis

3) Material properties for all materials including rubber, cord, and so on.

4) In numerical simulation, I do not see any detail explanation of your FE model. Please address all configuration, boundary condition, and element type, and so on related to FE model in detail.

5) In introduction, authors mentioned about analytical basis. I do not see any analytical basis related spring element model about air spring components. Related equations and analytical background formulation should be addressed.

Author Response

Thanks for the suggestion.Please see the attachment.

Reviewer 2 Report

Interesting work, combines numerical calculations with model verification. Very valuable research.

My comments below:

1. Poor literature review because it includes only 17 items. Literature review is not complete, missing some fundamental positions on numerical methods and justification for choosing finite element method, justification for choosing Abaqus for calculations.

2. Discussion chapter is missing in the paper. A paper without this chapter cannot be accepted for publication. You should compare the results obtained with existing ones, justify the correctness of the calculation methodology adopted, support the plan and idea of the experiment by citing that someone else has also already done research of this type.

The following is an article in which calculations were performed using the Abaqus program and it can be safely said that it is suitable for performing such computational simulations.

Staszak, N.; Garbowski, T.; Szymczak-Graczyk, A. Solid Truss to Shell Numerical Homogenization of Prefabricated Composite Slabs. Materials 2021, 14, 4120. https://doi.org/10.3390/ma14154120

You can use the article given below as a comparison to concrete aging and how the effect of concrete aging was modeled, i.e., concrete reaching higher compressive strength with time. Such calculations were carried out for example in the work:

Szymczak-Graczyk, A.; Ksit, B.; Laks I. Operational problems in structural nodes of reinforced concrete constructions. IOP Conference Series: Materials Science and Engineering, 603 (2019), 032096, doi:10.1088/1757-899X/603/3/032096.

Author Response

(The authors gave the same response as above.)

Round 2

Reviewer 1 Report

I am okay with authors' comments about my questions.

Reviewer 2 Report

Thank you for making the changes to the manuscript.